# Co-Expression Analysis Reveals Differential Expression of Homologous Genes Associated with Specific Terpenoid Biosynthesis in *Rehmannia glutinosa*

**DOI:** 10.3390/genes13061092

**Published:** 2022-06-19

**Authors:** Ji-Nam Kang, Jong-Won Han, So-Hee Yang, Si-Myung Lee

**Affiliations:** 1Genomics Division, National Institute of Agricultural Sciences, Jeonju-si 54874, Korea; greatnami@korea.kr (J.-N.K.); eumnada@korea.kr (S.-H.Y.); 2Herbal Crop Research Division, National Institute of Horticultural and Herbal Science, Eumseong 27709, Korea; pvphan@korea.kr

**Keywords:** Illumina, PacBio, terpenoid biosynthesis, sterol, saponin, carotenoid, iridoid

## Abstract

Terpenoids are naturally occurring compounds involved in respiration, photosynthesis, membrane fluidity, and pathogen interactions and are classified according to the structure of their carbon skeleton. Although most terpenoids possess pharmacological activity, knowledge about terpenoid metabolism in medicinal plants is insufficient. *Rehmannia glutinosa* (*R. glutinosa*) is a traditional herb that is widely used in East Asia and has been reported to contain various terpenoids. In this study, we performed a comprehensive transcriptome analysis of terpenoid metabolism in *R. glutinosa* using two RNA sequencing platforms: Illumina and PacBio. The results show that the sterol, saponin, iridoid, and carotenoid pathways are active in *R. glutinosa*. Sterol and saponin biosynthesis were mevalonate pathway dependent, whereas iridoid and carotenoid biosynthesis were methylerythritol 4-phosphate pathway dependent. In addition, we found that the homologous genes of key enzymes involved in terpenoid metabolism were expressed differentially and that the differential expression of these genes was associated with specific terpenoid biosynthesis. The different expression of homologous genes encoding acetyl-CoA acetyltransferase, 3-hydroxy-3-methylglutaryl-CoA reductase, mevalonate kinase, mevalonate diphosphate decarboxylase, farnesyl pyrophosphate synthase, squalene synthase, and squalene epoxidase was associated with sterol and saponin biosynthesis. Homologous genes encoding 1-deoxy-D-xylulose 5-phosphate synthase were also differentially expressed and were associated with carotenoid and iridoid biosynthesis. These results suggest that the biosynthesis of specific terpenoids can be regulated by the homologous of key enzymes involved in plant terpenoid metabolism.

## 1. Introduction

Terpenoids (also known as isoprenoids) are a large class of natural products that are widely distributed in higher plants [1,2]. Most terpenoids have pharmacological properties such as stress relieving, sleep inducing, anti-inflammatory, antispasmodic, and anti-cancer activity [1,2,3,4]. Additionally, terpenoids exhibit anti-microbial activity against Gram-positive/negative pathogenic bacteria [3] and anti-viral activity against coronaviruses that cause respiratory syndromes [5]. Terpenoids exist in myriad structurally diverse forms. Primary and secondary terpenoid metabolites in plants regulate respiration, photosynthesis, growth, and membrane fluidity and contribute to plant–pathogen and allelopathic interactions, respectively [2]. Sterols, ubiquinones, chlorophyll, and carotenoids are recognized as primary terpenoid metabolites, whereas artemisinin, taxol, saponins, and iridoids are known secondary terpenoid metabolites in plants [2,4].

The terpenoid pathway is divided into three steps. The first step involves the synthesis of 5-carbon (C5) isoprenoids, isopentenyl pyrophosphate (IPP), and dimethylallyl pyrophosphate (DAMPP), which are common terpenoid precursors [6]. IPP and DAMPP are produced by two independent pathways, mevalonate phosphate (MVA) and methylerythritol 4-phosphate (MEP) in plants [4,7]. The MVA pathway consists of the following enzymes: acetyl-CoA acetyltransferase (AACT), hydroxymethylglutaryl-CoA synthase (HMGS), 3-hydroxy-3-methylglutaryl-CoA reductase (HMGR), mevalonate kinase (MK), phosphomevalonate kinase (PMK), and mevalonate diphosphate decarboxylase (MVD), which are distributed in the cytoplasm, endoplasmic reticulum, and peroxisome [1]. IPP and DAMPP generated from the MVA pathway are primarily used for triterpenoid, sesquiterpenoid, and ubiquinone synthesis [2]. In contrast, the MEP pathway is localized in the plastid and involves sequential catalysis by the following enzymes: 1-deoxy-D-xylulose 5-phosphate synthase (DXS), 1-deoxy-D-xylulose 5-phosphate reductoisomerase (DXR), 2-C-methyl-D-erythritol 4-phosphate cytidylyltransferase (MCT), 4-diphosphocytidyl-2-c-methyl-d-erythritol kinase (CMK), 2-C-methyl-D-erythritol 2,4-cyclodiphosphate synthase (MCS), 4-hydroxy-3-methylbut-2-enyl diphosphate synthase (HDS), and 4-hydroxy-3-methylbut-2-enyl diphosphate reductase (HDR) [1]. IPP and DAMPP derived from the MEP pathway are primarily used as precursors for compounds, such as tetraterpenoids, monoterpenoids, chlorophyll, and plastoquinones [2]. Isopentenyl-diphosphate delta-isomerase (IDI) catalyzes the reversible conversion of IPP and DAMPP in the plastid and cytoplasm [1,7]. Both IPP and DAMPP are directly generated by HDR activity via the MEP pathway, whereas in the MVA pathway, DAMPP is generated by IDI activity from IPP [2,8].

The second step is the synthesis of the prenyl diphosphate precursor by terpene synthase (TPS) using C5 isoprenoid [9]. Prenyltransferase (PT) and terpene cyclase (TPC) are classified as TPS, and they contribute to the enormous diversity of terpenoids in plants [6,9,10]. PTs catalyze prenyl chain extensions [11]; geranyl pyrophosphate synthase (GPPS) catalyzes the formation of geranyl diphosphate (GPP, C10) by condensing one molecule each of DAMPP (C5) and IPP (C5); geranylgeranyl pyrophosphate synthase (GGPPS) catalyzes the condensation of three IPP molecules from DAMPP (C5) to form geranylgeranyl diphosphate (GGPP, C20); and farnesyl pyrophosphate synthase (FPPS) catalyzes the condensation of two molecules of IPP (C5) from DAMPP (C5) to form farnesyl diphosphate (FPP; C15) [2,11]. These PTs are responsible for the head-to-tail condensation reaction of IPP from DAMPP [1,7,11]. Furthermore, squalane synthase (SQS) and phytoene synthase (PSY) catalyze the head-to-head linkage of FPP and GGPP to form squalene (C30) and phytoene (C40), respectively [2,7,11]. Geraniol synthase (GES) is classified as a TPC and catalyzes the formation of geraniol from GPP [12].

The final step in the pathway is the post-transformation of specific prenyl diphosphate precursors through methylation, hydroxylation, isomerization, and glycosylation [1,4,13]. Triterpenoids are C30-type terpenoids composed of six isoprene units [2,3] and are commonly synthesized via the MVA pathway [4]. Phytosterols and saponins are triterpenoids that are ubiquitous in higher plants [3]. 2,3-oxidosqualene is synthesized by squalene epoxidase (SQE) from squalene, which is a common precursor of sterol and saponin pathways [2,4,7]. Cycloartenol synthase (CAS) catalyzes the formation of cycloartenol, a precursor of beta-sitosterol, from 2,3-oxidosqualene, and beta-amyrin synthase (BAS) converts 2,3-oxidosqualene to beta-amyrin, a precursor of saponin [4]. Higher plants form various sterol-based compounds from cycloartenol, which accumulate as glycosides or esters. In contrast, sapogenins, such as oleanolic acid, are formed from beta-amyrin by the oxidative activity of cytochrome P450 (CYP450), and saponin, a glycoside, is formed by uridine diphosphate-dependent glycosyltransferase (UGT) in plants [4].

Tetraterpenoids are C40-type terpenoids composed of eight isoprene units [2,3] and consist of carotenoids such as beta-carotene, lycopene, lutein, and zeaxanthin [3,13,14]. Carotenoids are usually synthesized via the plastid MEP pathway [14]. Lycopene is synthesized from the precursor phytoene by the sequential activation of phytoene dehydrogenase (PDS), 15-cis-zeta-carotene isomerase (ZIOS), zeta-carotene desaturase (ZDS), and carotenoid isomerase (CRTISO) [14]. Lycopene modification is an important step in generating carotenoid diversity and is catalyzed by lycopene epsilon cyclase (LCYE) and lycopene beta-cyclase (LCYB) [14]. Cyclization of one end of lycopene by LCYE directs the carotenoid flux toward alpha-carotene and lutein synthesis, whereas the cyclization of the two ends of lycopene by LCYB forms beta-carotene. Neoxanthin, the final product of the core-carotenoid biosynthesis pathway, is formed from beta-carotene by the sequential activation of beta-carotene 3-hydroxylase (BCH), zeaxanthin epoxidase (ZEP), and neoxanthin synthase (NXS) [14]. Neoxanthin is further modified to xanthoxin, the abscisic acid (ABA) biosynthetic precursor, by carotenoid cleavage dioxygenase (NCED) [14].

Monoterpenoids are C10-type terpenoids composed of two isoprene units that are primarily synthesized via the MEP pathway [2,3,7]. Iridoids are a large group of monoterpenoids classified as iridoid glycosides, simple iridoids, or non-glycosylated secoiridoids and bis-iridoids [3]. Two iridoid pathways have been identified in plants: the secoiridoid pathway for secologanin synthesis and iridoid glycoside pathway for catalpol synthesis. Iridotrial is formed from the precursor geraniol by the sequential activation of geraniol 8-hydroxylase (G8H), 8-hydroxygeraniol dehydrogenase (8HGO), iridoid synthase (IRS), and iridoid oxidase (IO); this is a common step in secologanin and catalpol synthesis [13,15]. The secologanin pathway is achieved by the sequential activation of deoxyloganetic acid glucosyltransferase (DLGT), deoxyloganic acid hydroxylase (DLH), loganic acid O-methyltransferase (LAMT), and secologanin synthase (SLS) [16]. The iridoid glycoside pathway is only partially known [13]; however, seven enzymes, aldehyde dehydrogenase (ALDH), flavanone 3-hydroxylase (F3H), 2-hydroxyisoflavanone dehydratase (2FHD), deacetoxycephalosporin-C hydroxylase (DCH), uroporphyrinogen decarboxylase (UPD), UDP-glucuronic acid decarboxylase (UGD), and squalene monooxygenase (SQM), catalyzing catalpol synthesis from iridotrial have been proposed in *Picrorhiza kurroa* [13,15]. A schematic map of extensive plant terpenoid metabolism is presented in Figure 1.

*Rehmannia glutinosa* (*R. glutinosa*), belonging to the Orobanchaceae family [17], is a traditional herb widely used in East Asia, including Japan, China, and Korea [18], and is abundant in primary and secondary terpenoids. Iridoids, beta-sitosterol, and carotene are the key active substances in this species [7,19,20]. However, their low yield in nature limits the bulk purification process and novel drug development [1,3,4]. Numerous studies have reported that the active components of *R. glutinosa* have a wide range of pharmacological effects on the blood, immune, endocrine, cardiovascular, and nervous system [18]. Therefore, a comprehensive understanding of terpenoid metabolism is essential for the development of new drugs by producing plants with high terpenoid content via plant metabolic engineering.

In this study, we comprehensively analyzed the expression of genes involved in terpenoid metabolism using two RNA sequencing (RNA-Seq) platforms: Illumina and PacBio. The co-expression analysis revealed the relative contributions of MVA and MEP pathways to specific terpenoid biosynthesis at the transcriptional level. In addition, homologous genes encoding key enzymes in terpenoid metabolism showed differential expression, associated with specific terpenoid biosynthesis. This study offers novel insights into plant metabolic engineering for the production of terpenoids.

## 2. Results

### 2.1. Construction of Unigene Set and Functional Annotations

The de novo assembly of transcripts is essential for transcriptome analysis for organisms with unclear reference genome. Therefore, we integrated Illumina paired-end and PacBio sequencing to generate a high-quality (HQ) unigene set for *R. glutinosa*. Raw reads ranging from 34,425,552 to 50,778,928 were generated by Illumina sequencing from 18 individual samples from *R. glutinosa* leaves, stems, flowers, and roots. The cleaned reads ranging from 32,234,332 to 48,176,356 were obtained after quality trimming (Appendix A). PacBio sequencing was performed to obtain full-length transcripts; 55,466 circular consensus subreads (CCSs) with an average length of 3095 bp were generated in total, among which, 55,093 CCSs were identified as HQ isoforms (Appendix A). The reads generated from the two sequencing platforms were combinedly assembled using rnaSPAdes, and sequences with >90% homology were removed using CD-HIT-EST to obtain the final 140,335 unigenes, with an average length of 1065 bp, N50 of 1821 bp, and a GC ratio of 39.72%. The BUSCO result using the Embryophyta odb10 database showed 98% coverage, indicating that the constructed *R. glutinosa* unigene set was created with HQ. Among 140,335 unigenes, 58,949 contained a protein-coding region (CDS) (Table 1).

Gene functions of the unigene set were predicted using the NCBI nr protein, gene ontology (GO), Kyoto Encyclopedia of Genes and Genomes (KEGG), InterProScan, and Araport11 databases. Of 140,335 unigenes, 77,747 had hits in the NCBI nr protein database. Additionally, 51,296, 45,663, 30,226, and 61,242 hits were found in GO, InterProScan, KEGG, and Araport11, respectively. A total of 78,559 unigenes had hits in all these databases, accounting for 55.98% of the total unigenes (Table 2).

Illumina RNA-Seq reads were mapped to the constructed unigenes in the range of 60.73–84.04%, with an average mapping rate of 76.27% (Appendix A). The expression levels of unigenes were calculated using transcripts per million (TPM) and the expression values for 140,355 unigenes are indicated in Appendix A along with annotation information.

### 2.2. Analysis of GO and KEGG Terms in Differentially Expressed Genes (DEGs)

Since *R. glutinosa* roots are the main source of medicinal components [18], we constructed seven DEG comparison combinations of root samples: (a) 60 DAP (days after planting) roots vs. 90 DAP roots; (b) 60 DAP roots vs. 120 DAP roots; (c) 90 DAP roots vs. 120 DAP roots; (d) 90 DAP roots vs. 90 DAP leaves; (e) 90 DAP roots vs. 90 DAP flowers; and (f) 90 DAP roots vs. 90 DAP stems. Up- and down-regulated DEGs in each comparative combination are shown in red and blue, respectively (Figure 2A). Terpenoid-related GO and KEGG terms were detected using stringent filtering (*p*-value < 0.01 and FDR < 0.01). Twenty GO terms, including tetraterpenoid, terpenoid, sterol, steroid, isoprenoid, IPP, and carotenoid biosynthesis, have been identified in biological processes of up-regulated GO terms. Terpenoid and isoprenoid metabolism-related GO terms (GO:0006721, GO:0016114, GO:0006720, and GO:0008299) were active in all comparisons. The IPP biosynthetic-related GO terms (GO:0046490, GO:0019287, and GO:0009240) were specifically activated in comparison-b. Steroid metabolism GO terms (GO:0008202 and GO:0006694) were active in all combinations except for comparison-d. Sterol-related GO terms (GO:0016125, GO:0016126, GO:0016128, GO:0016129, GO:0008204, and GO:0006696) showed predominant activity in comparisons-b and -c. The predominant activities of tetraterpenoid and carotenoid-related GO terms (GO:0016108, GO:0016109, GO:0016116, and GO:0016117) were confirmed in comparisons-d and -e (Figure 2B). Activities of these 20 GO terms were not confirmed in comparison-f (Appendix A).

Six terpenoid-related KEGG terms were identified in up-regulated KEGG terms. The activities of ubiquinone and other terpenoid-quinone biosynthesis pathways (KO00130) were confirmed in all the comparative combinations. The activity of the terpenoid backbone biosynthesis pathway (KO00900) was observed in all the combinations except comparison-f. Steroid biosynthesis (KO00100) and the sesquiterpenoid and triterpenoid biosynthesis pathways (KO00909) were activated in comparisons-b and -c. The monoterpenoid biosynthetic pathway (KO00902) is mainly active in comparisons-d and -e. The activity of the carotenoid biosynthetic pathway (KO00906) was detected in comparisons-a, -b, -d, and -e (Figure 2C). These results indicate that various terpenoid biosynthetic pathways were active in *R. glutinosa*. Triterpenoid biosynthesis, including sterols, is predicted to be active in roots during developmental phase and flowers, while monoterpenoid biosynthesis is expected to mainly be active in leaves and flowers. Carotenoid biosynthesis is thought to mainly be active in roots during developmental phase, flowers, and leaves. Additional information regarding the GO and KEGG terms for each comparative combination is provided in Appendix A.

### 2.3. Expression Analysis of Genes Involved in MVA and MEP Pathways

GO and KEGG analyses results suggest that terpenoid metabolism is activated in *R. glutinosa*. The expression of 61 genes involved in the MVA and MEP pathways was detected in the *R. glutinosa* transcriptome. Among them, 26 genes encoding AACT, HGMS, HGMR, MK, PMK, and MVD in the MVA pathway and 32 genes encoding DXS, DXR, MCT, CMK, MCS, HDS, and HDR in the MEP pathway were identified. Additionally, the expression of three IDI-encoding genes was also detected (Figure 3A and Appendix A). Co-expression analysis revealed that these genes could be classified into three groups based on their expression patterns (Figure 3B).

Cluster-a showed a strong expression in 120 DAP roots. Two AACT-, one HMGS-, five HMGR-, one MK-, and five MVD-encoding genes involved in the MVA pathway were included in cluster-a, along with one IDI-encoding gene (Figure 3B(a)).

Cluster-b was constitutively expressed in most tissues, except for 60 DAP roots and 90 DAP leaves and was relatively highly expressed in the flowers. Four AACT-, two HMGS-, one HGMR-, one MK-, two PMK-, and one MVD-encoding gene comprising the MVA pathway were identified in cluster-b (Figure 3B(b)).

Cluster-c was strongly expressed in the leaves and flowers at 90 DAP. Two DXS-, one DXR-, two MCT-, three CMK-, three MCS-, three HDS-, and five HDR-encoding genes comprising the MEP pathway were identified in cluster-c, along with two IDI-encoding genes (Figure 3B(c)).

These results indicate that the MVA pathway is primarily active in the roots and flowers, whereas the MEP pathway is predominantly active in the leaves and flowers. Interestingly, most genes involved in the MVA pathway were separated into two groups, indicating that the homologous genes of the major enzymes in the MVA pathway were differentially expressed. Similarly, the homologous genes encoding IDI were also differentially expressed in the two groups.

### 2.4. Specific Co-Expression of TPS-Encoding Genes

The enzymes, FPPS, GGPPS, GPPS, GGR, GES, SQS, and PSY, involved in the synthesis of specific terpenoid precursors in plants are classified as TPS [6,7,9]. We confirmed the expression of 23 genes encoding these enzymes in the *R. glutinosa* transcriptome (Figure 4A and Appendix A). Co-expression analysis showed that these genes could be classified into three groups based on their expression patterns.

Cluster-a showed specific expression in 120 DAP roots and contained four FPPS-encoding genes and six SQS-encoding genes (Figure 4B(a)).

Cluster-b showed high expression, primarily in 90 DAP flowers. Two FPPS-, one SQS-, three GGPPS-, two GPPS-, two GGR-, and one PSY-encoding genes were identified in cluster-b (Figure 4B(b)).

Cluster-c was specifically expressed in 90 DAP leaves and contained one GGR- and one GES-encoding gene (Figure 4B(c)).

The distinct co-expression of genes encoding FPPS and SQS suggests a functional link between these genes and squalane synthesis. The leaves-specific expression of GGR- and GES-encoding genes indicates the potential for the co-regulation of these genes for geraniol production. In cluster-b, the expression of the PSY-encoding gene was similar to that of the two GGPPS-encoding genes, suggesting that a functional link between GGPPS and PSY might play a role in phytoene production.

### 2.5. Expression Analysis of Genes Involved in Sterol and Saponin Biosynthesis Pathways

GO and KEGG analyses revealed that triterpenoid biosynthesis is active in *R. glutinosa*. We examined the *R. glutinosa* transcriptome for genes involved in saponin and sterol biosynthesis. The expression of six SQE-encoding genes that catalyze the formation of 2,3-oxidosqualene from squalene and three BAS- and seven BAO/CYP716A-encoding genes involved in saponin biosynthesis was detected. The expression of two CAS-, five SMT1-, and two SMT2-encoding genes involved in beta-sitosterol biosynthesis was also confirmed (Figure 5A and Appendix A). Co-expression analysis, including SQS-encoding genes, revealed that genes involved in sterol and saponin biosynthesis can be divided into two groups based on their expression patterns.

Cluster-a primarily contained the genes that were strongly expressed in 120 DAP roots. Three BAS-encoding genes and five BAO/CYP716A-encoding genes involved in saponin biosynthesis were included in this group, along with six SQS-encoding genes and one SQE-encoding gene (Figure 5B(a)).

Cluster-b consisted of genes that were constitutively expressed in all tissues except 60 DAP roots and 90 DAP leaves; however, the expression was relatively high in 90 DAP flowers. One CAS-, three SMT1-, and two SMT2-encoding genes involved in beta-sitosterol synthesis were co-expressed in cluster-b with one SQS- and five SQE-encoding genes (Figure 5B(b)).

These results indicate that both the sterol and saponin synthesis pathways are activated in *R. glutinosa*. Saponin was inductively synthesized in 120 DAP roots, while beta-sitosterol was likely synthesized constitutively in most tissues, except for 60 DAP roots and 90 DAP leaves. Interestingly, differential expression among homologous genes encoding SQS and SQE was associated with saponin and beta-sitosterol biosynthesis genes, respectively (Figure 5B).

Lanosterol synthase (LAS) catalyzes the production of lanosterol from 2,3-oxidosqualene via the sterol biosynthetic pathway [4]. We identified eight genes encoding LAS in the transcriptome of *R. glutinosa*; however, their expression levels were very low (TPM < 5) (Appendix A). These results suggest that sterol synthesis flux in *R. glutinosa* is directed toward beta-sitosterol rather than lanosterol.

### 2.6. Expression Analysis of Genes Responsible for the Carotenoid Pathway

The GO and KEGG analyses revealed the activation of the carotenoid pathway in *R. glutinosa*. A total of 49 carotenoid biosynthesis-related genes, including PDS, ZISO, ZDS, CRTISO, LCYB, LCYE, lutein deficient 5 (LUT5), lutein deficient 1 (LUT1), BCH, ZEP, NCED, and xanthoxin dehydrogenase (ABA2), were identified (Figure 6A and Appendix A). The co-expression analysis showed that most of the genes involved in carotenoid biosynthesis were mainly expressed in leaves and flowers (Figure 6B).

Genes encoding PDS, ZISO, ZDS, CRTISO, LCYB, LUT5, LUT5, BCH, ZEP, NCED, and ABA2 were co-expressed in 90 DAG flowers along with genes encoding PSY (Figure 6B(b)). Moreover, the high expression of one LCYE- and two LCYB-encoding genes was confirmed in 90 DAP leaves and flowers (Figure 6B(c)).

In carotenoid metabolism, neoxanthin and capsanthin formation from violaxanthin are catalyzed by NXS and capsanthin–capsorubin synthase (CCS), respectively [14]. However, the expression of the genes encoding these enzymes was very low (TPM < 5) in the *R. glutinosa* transcriptome (Appendix A), and two ABA2 encoding genes showed basal expression (Appendix A). In contrast, the highest expression level of the BCH-encoding gene *Rg_038580* was detected in the 90 DAP flowers (Appendix A). These results suggest that the carotenoid metabolism was activated with a primary focus on zeaxanthin production in the 90 DAP flowers of *R. glutinosa*.

### 2.7. Identification and Expression Analysis of Iridoid Biosynthesis Genes

GO and KEGG analyses revealed the potential of monoterpenoid synthesis in *R. glutinosa*. Iridoids are a large group of monoterpenoids, which are reportedly abundant in *R. glutinosa* [19]. From these transcriptome data, we identified the expression of 15 genes encoding G8H, 8HGO, and IRS which are involved in the common enzymatic steps of the two iridoid pathways. The expression of four DLGT- and four SLS-encoding genes responsible for secologanin biosynthesis was also identified. Furthermore, the expression of 12 genes encoding F3H, 2HFD, UPD/UGD, and SQM, which are presumed to be catalpol biosynthesis genes, was also detected (Figure 7A and Appendix A).

The expression analysis of genes involved in the common enzymatic step showed that genes encoding GES, 8HGO, and IRS were co-expressed in 90 DAP leaves (Figure 7B(c)). Additionally, genes encoding DLGT and SLS, which are involved in secologanin biosynthesis, were also co-expressed in the 90 DAP leaves with genes encoding GES, 8HGO, and IRS (Figure 7C(c)). These results suggest the potential for secologanin production in *R. glutinosa*. However, several genes responsible for secologanin biosynthesis were missing from the *R. glutinosa* transcriptome. The expression of genes encoding IO, DLH, and LAMT was not detected, and the expression of genes encoding G8H was distinct from that of the genes involved in the common enzymatic step of the iridoid pathway (Figure 7B). It has been reported that G8H, IO, and DLH all belong to the CYP450 superfamily [16]. Therefore, we performed the co-expression analysis of 74 CYP450-encoding genes, which were significantly higher in 90 DAP leaves than in 90 DAP roots, with GES, 8HGO, IRS, DLGT, and SLS encoding genes (Appendix A). Fifteen CYP450-encoding genes displayed similar expression patterns to those encoding GES, 8HGO, IRS, DLGT, and SLS; most of these CYP450-encoding genes were annotated as CYP72 and CYP76 (Appendix A).

The expression of putative catalpol biosynthesis genes was distinct from that of the genes involved in the common steps of the iridoid pathway. As shown in Figure 7D, most of the genes encoding F3H and SQM belonged to cluster-b with a strong expression in 90 DAP flowers. Two 2HFD-encoding genes with a high expression in 120 DAP roots were included in cluster-a. The three UPD-encoding genes belonged to cluster-c along with the GES-, 8HGO-, and IRS-encoding genes, although the UPD-encoding genes showed a similar expression in both 90 DAP leaves and flowers (Figure 7D). This result suggests that the previously reported enzymes involved in catalpol biosynthesis may not be applicable in *R. glutinosa*.

### 2.8. Comprehensive Co-Expression Analysis of Genes Involved in Terpenoid Biosynthesis

We systematically investigated the expression of a large number of genes involved in specific terpenoid biosynthesis, including the MVA and MEP pathways. The co-expression of these genes may suggest a functional link for specific terpenoid biosynthesis. Therefore, the integrated co-expression analysis was performed using 186 genes selected from the previous results. These genes were classified into four groups based on their expression patterns (Figure 8 and Appendix A).

Cluster-a consisted of genes that were highly expressed genes in 120 DAP roots. Genes encoding AACT, HMGS, HMGR, MK, and MVD, which constitute the MVA pathway, were identified in this group. In addition, the genes encoding FPPS, SQS, SQE, BAS, and BAO were included in cluster-a (Figure 8a). This result suggests that saponin can be synthesized through the MVA pathway and FPPS activity in the 120 DAP roots of *R. glutinosa*.

Cluster-c consisted of genes showing a high expression in 90 DAP leaves. Genes encoding DXS, DXR, MCT, CMK, MCS, HDS, and HDR involved in the MEP pathway and one GGR-encoding gene were included in cluster-c. Additionally, genes encoding GES, 8HGO, IRS, DLGT, and SLS also belonged to this group (Figure 8c). This result indicates the potential for secologanin synthesis via the MEP pathway and GGR activity in 90 DAG leaves of *R. glutinosa*.

Cluster-b consisted of genes that were strongly expressed in 90 DAP flowers. Genes encoding DXS, DXR, CMK, HDS, and HDR, which are involved in the MEP pathway, belonged to this group. In addition, genes encoding PSY, PDS, ZISO, ZDS, CRTISO, LCYB, LCYE, BCH, LUT5, LUT1, ZEP, and NCED were also classified in cluster-b (Figure 8b). This result implies that carotenoids can be produced via the MEP pathway in 90 DAG flowers of *R. glutinosa*.

Cluster-d genes showed a constitutive expression in all tissues, except for 60 DAP roots and 90 DAP leaves. Genes encoding AACT, HMGR, MK, PMK, and MVD involved in the MVA pathway were included in cluster-d along with genes encoding SQS, SQE, CAS, SMT1, and SMT2 (Figure 8d). This result suggests that sterol synthesis mainly occurs through the MVA pathway and is active in the roots’ development, flowers, and stems of *R. glutinosa*.

Collectively, these results imply that saponins and sterols are synthesized via the MVA pathway and that iridoids and carotenoids are produced via the MEP pathway in *R. glutinosa*. Furthermore, in this process, homologous genes encoding major enzymes involved in terpenoid biosynthesis showed differential expression patterns and were co-expressed with genes involved in specific terpenoid biosynthesis. Therefore, these results suggest that the biosynthesis of specific terpenoids may be regulated by the homologous of the key enzymes involved in terpenoid metabolism.

### 2.9. Quantitative Real-Time Polymerase Chain Reaction (qRT-PCR) Test of Homologous Genes Encoding Major Enzymes in Terpenoid Metabolism

This study revealed the possibility that the differential expression of homologous genes encoding key enzymes in terpenoid metabolism could contribute to specific terpenoid biosynthesis at the transcriptional level. Therefore, we performed qRT-PCR analysis to confirm the real expression of key homologous genes of terpenoid metabolism. Leaf, stem, and flower tissues from 120 DAP plants that were excluded from RNA-Seq analyses were used for qRT-PCR analysis along with the RNA samples from 90 DAP plants. Gene structure analysis was performed to design specific primers for target gene amplification (Figure 9A and Figure 10A).

Based on the co-expression profiles shown in Figure 8, the Rg_028863 (AACT), Rg_015359 (HGMR), Rg_031322 (MK), Rg_029638 (MVD), Rg_030059 (FPPS), Rg_023409 (SQS), Rg_016154 (SQE), Rg_008321 (CAS), and Rg_029802 (SMT1) genes are expected to be involved in sterol biosynthesis through the MVA pathway, whereas the Rg_023705 (AACT), Rg_016308 (HGMR), Rg_019629 (MK), Rg_025679 (MVD), Rg_033701 (FPPS), Rg_064360 (SQS), Rg_021808 (SQE), Rg_012151 (BAS), and Rg_022544 (BAO) genes are predicted to be associated with saponin biosynthesis via the MVA pathway (Figure 8 and Appendix A). The qRT-PCR results clearly showed differential expression patterns between the homologous genes. The expression of genes predicted to be involved in sterol biosynthesis was mostly similar or lower in 120 DAP plants than in 90 DAP plants. In contrast, the expression of genes predicted to function in saponin biosynthesis was significantly induced in most tissues of 120 DAP plants compared to those of 90 DAP plants (Figure 9B).

Furthermore, we investigated the real expression of genes involved in carotenoid and iridoid biosynthesis via the MEP pathway. Based on the co-expression data shown in Figure 8, the genes *Rg_011504* (*DXS*), *Rg_021081* (*DXR*), *Rg_026395* (*CMK*), *Rg_022659* (*HDS*), *Rg_027939* (*HDR*), *Rg_031117* (*GGPPS*), *Rg_020168* (*PSY*), *Rg_010306* (*PDS*), *Rg_013137* (*ZDS*), *Rg_019608* (*LCYB*), and *Rg_038580* (*BCH*) were considered to play a role in carotenoid biosynthesis, whereas, *Rg_013388* (*DXS*), *Rg_036125* (*DXR*), *Rg_044226* (*MCT*), *Rg_027961* (*CMK*), *Rg_044118* (*MCS*), *Rg_012778* (*HDS*), *Rg_024053* (*HDR*), *Rg_027010* (*GGR*), *Rg_022299* (*GES*), *Rg_024869* (*8HGO*), *Rg_036069* (*IRS*), *Rg_023087* (*DLGT*), and *Rg_026493* (*SLS*) were involved in iridoid production (Figure 8 and Appendix A). qRT-PCR results demonstrated that the expression pattern of the two homologous genes encoding DXS was clearly distinct. The *Rg_011504* (*DXS*) gene was highly expressed in the flowers of 90 DAP and 120 DAP plants, and this expression pattern was similar to that of the genes involved in carotenoid biosynthesis. In contrast, the *Rg_013388* (*DXS*) gene showed a strong expression in the leaves of 90 DAP and 120 DAP plants, similar to the expression of genes involved in iridoid biosynthesis (Figure 10B).

Additionally, we performed qRT-PCR analysis on three homologous genes encoding IDI. The expression of the *Rg_009212* gene was induced in 120 DAP plants compared to that in 90 DAP plants, which is similar to the expression pattern of genes involved in saponin biosynthesis via the MVA pathway. *Rg_037230* showed a high expression in 90 DAP and 120 DAP leaves and flowers and showed a similar expression to the genes constituting the MEP pathway (Appendix A).

These results support the RNA-Seq conclusion that the homologous genes of key enzymes involved in terpenoid metabolism are differentially expressed and that this differential expression is associated with the expression of genes involved in specific terpenoid biosynthesis. Additional qRT-PCR results for the genes involved in terpenoid metabolism are shown in Appendix A. The nucleotide sequences and primer information for the analyzed genes are shown in Appendix A.

## 3. Discussion

### 3.1. Number of Abundant Transcripts in R. glutinosa

PacBio sequencing is a full-length transcriptome sequencing technology that can detect splicing events in genes [21,22]. This technique provides longer reads than other NGS platforms and requires no assembly; however, it has a higher sequencing error rate. In contrast, Illumina short-read sequencing provides sufficient depth with a low error rate; however, it is difficult to recover full-length transcripts due to the complexity of alternative splicing mechanisms during assembly [21,23]. To overcome this problem, we generated individual reads using both sequencing methods, namely PacBio and Illumina, and performed hybrid assembly using rnaSPAdes [23].

As a result, 140,335 unigenes were identified from *R. glutinosa*. In *Arabidopsis*, it is estimated that an average of 300,000 transcripts are derived from approximately 25,000 genes [21]. These transcripts in plants may contain protein coding sequences, isoforms, and non-coding RNAs [21,22,24]. A reference genome for *R. glutinosa* has recently been reported and is predicted to contain 48,475 protein-coding genes [25]. The number of predicted genes differs from the 58,949 predicted CDS in our study (Table 1), possibly because our transcriptome data also contain the isoforms of genes generated by alternative splicing [13].

### 3.2. Differential Expression of Homologous Genes for Specific Terpenoid Biosynthesis

Triterpenoids are commonly produced via the MVA pathway in higher plants [2,4,26,27]. Our data indicated that triterpenoid biosynthesis, including sterol and saponin biosynthesis, was MVA pathway dependent. The genes involved in the MVA pathway were primarily expressed in roots, flowers, and stems; however, their expression was low in the leaves (Figure 3B). In mature *Arabidopsis*, MVA pathway genes are predominantly expressed in the roots, hypocotyl, flower, and seed [26]. Interestingly, we found that homologous genes encoding key enzymes in the MVA pathway were differentially expressed. Similar results were observed for FPPS-, SQS-, and SQE-encoding homologous genes (Figure 4 and Figure 5). The end product produced by the sequential activity of AACT to SQE is 2,3-oxidosqualene, which is a branch point between sterol and saponin metabolism [4]. Therefore, we predicted that the different expression patterns of these genes would be involved in saponin and sterol synthesis. The expression analysis of genes encoding CAS, SMT1, SMT2, BAS, and BAO confirmed our prediction (Figure 9B).

Several studies reported that the homologous genes involved in the MVA pathway and triterpenoid synthesis play other roles in the plants. There are two AACT paralogs in the *Arabidopsis* genome, among which only AACT2 is involved in sterol production and is essential for plant growth and development. Compared to the wild-type plants, AACT1 mutant plants did not display a growth inhibitory phenotype and were unable to displace AACT2 function. It has been suggested that the presence of AACT1 in the *Arabidopsis* genome might be related to other biological processes [28]. HGMR is important for terpenoid synthesis flux as a rate-limiting enzyme in the MVA pathway [29]. In *Panax ginseng*, HMGR1 is suggested to play a role in the synthesis of primary metabolites such as sterols, whereas HMGR2 is suggested to play a specific role in the age-dependent increase in saponin [29]. In *Solanum tuberosum*, differential expression patterns of the three HMGR homologous genes were associated with sterol and sesquiterpenoid phytoalexin biosynthesis, and it was proposed that separate MVA pools exist for sterol and sesquiterpenoid phytoalexin biosynthesis [30]. In *Arabidopsis*, FPPS has two isoforms with distinct expression patterns. Cunillera et al. (1996) suggested that the differential expression of *Arabidopsis* FPPS1 and FPPS2 may be due to the specificity of each FPPS isoform for the induction of specific isoprenoid products [31]. The distinct phenotypes of individual mutants of *Arabidopsis* FPPS1 and FPPS2 imply that these genes have different functions [32]. Among the three SQS homologous genes that are functionally expressed in *P. ginseng*, *PgSQS1* is abundantly expressed in all tissues and is involved in sterol and saponin biosynthesis. However, *PgSQS3* was preferentially expressed in leaves and was significantly induced by treatment with MeJA, a saponin synthesis inducer. Saponins accumulate more in the leaves than in the roots of *P. ginseng* [33]. In addition, the expression of two SQE homologous genes, namely PgSQE1 and PgSQE2, was differently regulated in *P. ginseng*. *PgSQE1* was abundantly expressed in most tissues and was significantly induced by MeJA treatment. In contrast, *PgSQE2* was only expressed in petioles and flowers and was strongly repressed by MeJA treatment. The expression of *PgSQE1* was suppressed in PgSQE1 RNA interference transformants, leading to a decrease in saponin content. Remarkably, the expression of the *PgSQE2* gene was stimulated in PgSQE1 RNA interference transformants, leading to sterol accumulation. The authors predicted that PgSQE1 regulates saponin biosynthesis and that PgSQE2 may be involved in sterol biosynthesis [34]. These results indicate that sterol and saponin biosynthesis via the MVA pathway is regulated by the homologous genes of the key enzymes.

Previous studies have revealed that carotenoids and monoterpenoids are primarily produced via the MEP pathway [2,14,26,27]. Complying with these studies, our data also showed that carotenoid and iridoid biosynthesis primarily occurred via the MEP pathway in *R. glutinosa*. Furthermore, the expression of genes constituting the MEP pathway was predominantly abundant in the *R. glutinosa* leaves and flower (Figure 3B). The predominant activity of the MEP pathway in photosynthetic tissues such as rosette leaves and flowers has been identified in mature *Arabidopsis* [26]. We found that homologous genes encoding several key enzymes in carotenoid and iridoid biosynthesis, via the MEP pathway, were differentially expressed. In particular, *Rg_011504* and *Rg_013388* genes encoding DXS showed tissue-specific differential expression, which were strongly associated with the expression patterns of PSY-and GES-encoding genes, respectively (Figure 10B). PSY and GES have been reported as precursors of the carotenoid and iridoid biosynthetic pathways, respectively [12,14,35]. Therefore, we investigated the expression of genes involved in carotenoid and iridoid biosynthesis in the *R. glutinosa* transcriptome. Key genes involved in carotenoid biosynthesis were abundantly expressed in flowers and co-expressed with *Rg_011504* (*DXS*) and *Rg_020168* (*PSY*) genes. Genes responsible for iridoid biosynthesis were predominantly co-expressed in the leaves with *Rg_013388* (*DXS*) and *Rg_022299* (*GES*) genes (Figure 10B). These results suggest the distinct role of DXS in carotenoid and iridoid biosynthesis in the MEP pathway.

DXS is considered to be a rate-limiting enzyme in the MEP pathway [36], and the specific regulation of DXS homologous genes for specific terpenoid synthesis has been reported in previous studies. In *Artemisia annua*, sesquiterpenoid artemisinin biosynthesis is specifically regulated by DXS2 [36]. In *Oryza sativa*, DXS3 is co-expressed with diterpenoid phytoalexin biosynthesis genes, and DXS2 regulates carotenoid synthesis in the seeds [37,38]. DXS and PSY are co-expressed to regulate carotenoid biosynthesis in *Solanum lycopersicum* [39]. Additionally, we confirmed that the GGR-encoding gene, *Rg_027010*, was co-expressed with genes involved in iridoid biosynthesis (Figure 10B). The GGR enzyme exhibits GPPS small subunit (GPPS-SSU) activity [7], and GPPS-SSU plays an important role in driving the terpenoid metabolic flux to monoterpenoid synthesis in *Catharanthus roseus*. GPPS-SSU is stimulated by biotic/abiotic stress, and MeJA treatment induces the genes involved in the MEP pathway such as DXS, DXR, MCS, and HDS, along with CrGPPS-SSU [40]. In contrast, GGPPS is co-expressed with genes involved in the MEP and carotenoid pathways [11]. DXS and GGR are entry-point enzymes involved in various terpenoid biosynthetic processes [40,41]. As the regulation of the pathways competing for the same precursor primarily occurs by the first enzyme of a competing branch [30], homologous DXS and GGR have the potential to play a key role in regulating carotenoid and monoterpenoid synthesis via the MEP pathway.

Since DAMPP is a starter unit for isoprenoid chain elongation, IDI activity is essential for DAMPP supply in terpenoid metabolism [2]. In the present study, three homologous genes encoding IDI were identified, and their expression patterns were determined. The expression of *Rg_009212* was presumed to play a role in DAMPP generation in the saponin biosynthesis process through the MVA pathway, whereas *Rg_037230* was expected to be involved in DAMPP generation from the MEP pathway (Appendix A). Several studies have explored the distinct functions of IDI homologous genes. In *Nicotiana tabacum*, IDI1 targets plastids, and IDI2 is found in the cytoplasm [8,42]. In *Arabidopsis*, IDI1 and IDI2 target plastids and mitochondria, respectively; however, the alternative splicing forms of both genes are found in the cytoplasm [8]. The different subcellular localizations of IDI proteins suggest that IDI homologous play distinct roles in the production of DAMPP in the MVA and MEP pathways.

The expression of genes encoding HGMS, PMK, DXR, MCT, CMK, HDS, and HDR did not show a distinct association with specific terpenoid synthesis in this study (Figure 8, Figure 9, Figure 10, Appendix A), thus suggesting that these genes have the potential to play common roles in the synthesis of various terpenoids. However, the possibility that these genes are specifically regulated for the synthesis of specific terpenoids should not be excluded. Ubiquinones, sterols, saponins, and sesquiterpenoids are a class of terpenoids synthesized primarily via the MVA pathway, and monoterpenoids, diterpenoids, carotenoids, gibberellins, tocopherols, chlorophyll, and plastoquinones are terpenoids commonly synthesized via the MEP pathway [1,26]. Further studies on these specific terpenoid synthesis pathways may provide additional information on homologous genes involved in terpenoid biosynthesis via the MVA/MEP pathway.

Our results suggest that specific terpenoid biosynthesis in the extensive plant terpenoid metabolism can be temporally and spatially regulated by the homologous genes of key enzymes. Homologous genes may be isoforms or paralogs generated by alternative splicing and gene duplication, respectively; these are evolutionary strategies in organisms to generate protein diversity [43]. Spliced isoforms can be translated into different proteins from the corresponding mRNAs and can exhibit different expression levels at different stages of development [21]. Paralogs are independently regulated and can exhibit spatial and temporal differential expression patterns [26]. Homologous genes may be important in promoting adaptation to dynamic environments in plants and may be involved in specific metabolic fluxes, reducing the need for metabolic crosstalk [26]. In this study, the differential expression of homologous genes in terpenoid metabolism is likely a plant strategy to meet the increased demand for specific terpenoids.

### 3.3. Activity of the Saponin Pathway in Triterpenoid Metabolism

Saponin has not yet been reported as a major bioactive substance in *R. glutinosa*. Therefore, the inducible expression of genes involved in saponin biosynthesis is an interesting result. Perhaps these genes were induced by environmental stress. Bacteria, viruses, and fungi are biological inducers of terpenoids [1]. Terpenoid biosynthesis occurs in response to plant growth and biotic and abiotic environmental factors, and is induced by pathogen attack and insect feeding [4]. In many plants, phytoalexin biosynthesis via the isoprenoid pathway is increased by fungal and bacterial infections, accompanied by increased AACT, HMGS, HMGR, MK, and MVD activity [44]. Saponins are isoprenoid phytoalexins that have been reported to exhibit resistance to pathogenic fungi and bacteria in several studies [4,45]. In *Avena sativa*, saponin avenacin biosynthesis genes are tightly co-regulated in the root epidermis, inducing resistance to phytopathogenic fungi [4]. The ‘Tokang’ cultivar used in this study is resistant to root rot [20]. Inducible saponin biosynthesis in *R. glutinosa* is probably a unique characteristic of the ‘Tokang’ variety that is possibly related to disease resistance.

### 3.4. Carotenoid Biosynthesis from Tetraterpenoid Metabolism

Several types of carotenoids have been identified in plants. Lycopene in tomato, beta-carotene in carrot, capsanthin in pepper, and lutein in marigold flowers are well-known carotenoids [14]. Our transcriptome data suggest that zeaxanthin production is a major source of carotenoids in *R. glutinosa* flowers. Thus far, studies on carotenoid biosynthesis in *R. glutinosa* are limited. However, the strong expression of the PSY-encoding gene provides compelling evidence for carotenoid synthesis in the *R. glutinosa* flowers (Figure 10B and Appendix A). Furthermore, carotenoids have been reported to be beneficial to human health owing to their antioxidant and anti-aging properties [14]. Additional studies on carotenoid metabolism may be an effective way to improve the nutritional quality of *R. glutinosa*.

### 3.5. Biosynthesis of Iridoids from Monoterpenoid Metabolism

The iridoid pathway consists of the secologanin and catalpol pathways; however, very limited information is available regarding the catalpol pathway in plants [13]. In this study, we analyzed the expression of genes involved in the secologanin and catalpol pathways based on the co-expression profiles of GES-, 8HGO-, and IRS-encoding genes involved in the common steps of iridoid synthesis. The results suggest that the secologanin pathway is active in *R. glutinosa*; however, the catalpol pathway was not confirmed in *R. glutinosa* (Figure 7), probably because of the limited knowledge of the catalpol pathway in plants. Enzyme activity prediction and transcriptome analysis in *P. kurroa* suggested that UGT, ALDH, F3D, 2HFD, DCH, UPD/UGD, and SQM are late enzymes in the catalpol biosynthetic pathway. However, the actual function of these enzymes in catalpol production has not yet been validated in plants [13,15]. Transcriptome data in our study showed that the expression of genes encoding F3H, 2HFD, and SQM was completely different from that of the common iridoid pathway genes (Figure 7D), suggesting that these genes are not involved in catalpol synthesis, at least in *R. glutinosa*. Further studies on other enzymes with similar functions are required to clarify the catalpol biosynthesis pathway in *R. glutinosa*, and enzymes with hydroxylase, dehydratase, decarboxylase, and monooxygenase activities may be suitable candidates for the catalpol pathway [15].

In addition, the genes encoding G8H, IO, DLH, and LAMT in the secologanin pathway were missing from our results. However, the expression analysis of genes encoding the CYP450 superfamily suggested that CYP72 and CYP76 encoding genes participate in secologanin biosynthesis in *R. glutinosa* (Appendix A and Appendix A). In *C. roseus*, CYP76B6, CYP76A26, and CYP72A224 function as G8H, IO, and DLH, respectively [16]. An alternative pathway was reported for secologanin synthesis, which may account for the missing expression of LAMT-encoding genes [46].

Our results show that the genes involved in the common pathway of iridoids were predominantly expressed in the leaves than in the roots of *R. glutinosa*. The GES, G10H, IRS, and IO genes are more abundantly expressed in the leaves than in the roots of *Centranthera grandiflora* [13]. These results indicate that catalpol is primarily synthesized in the leaves rather than in the roots. However, catalpol has been reported to accumulate in the roots of *R. glutinosa* [20,47,48], suggesting that catalpol is preferentially synthesized in the leaves and then transported to the roots for accumulation [13].

## 4. Conclusions

In this study, we comprehensively investigated the expression of genes involved in terpenoid metabolism using HQ transcriptome data in *R. glutinosa*. The results show the relative contributions of the MVA and MEP pathways to monoterpenoid, triterpenoid, and tetraterpenoid biosynthesis at the transcriptional level. In this process, we found that the homologous genes of the key enzymes involved in terpenoid metabolism were differently expressed and that this difference in expression was associated with specific terpenoid biosynthesis. This study provides additional information for the development of medicinal plants that produce and accumulate higher levels of terpenoids via metabolic engineering. RNA-Seq is a powerful tool for predicting the candidate genes involved in extensive plant terpenoid metabolism, and the current research community needs to comprehensively study the functions of novel genes involved in the biosynthesis of medicinally active compounds. Our future research will focus on the enzymatic activity of candidate genes and the analysis of terpenoid metabolites.

## 5. Materials and Methods

### 5.1. Plant Materials and RNA Preparation

*R. glutinosa* cultivar ‘Tokang’ was provided by the National Institute of Horticulture and Herbal Science, Korea. The rhizomes of *R. glutinosa* stored in an ice chamber (maintained at 1 °C) were grown for three months after transplantation into the field. Leaves, stems, and roots of *R. glutinosa* were collected 60 DAP. The second sampling of the leaves, stems, roots, and flower was performed at the flowering stage 90 DAP. Lastly, leaf, stem, root, and flower samples were collected 120 DAP. All samples were individually collected from three independent plants as biological replicates and immediately frozen by liquid nitrogen. Subsequently, RNA was extracted from each sample using the cetyltrimethylammonium bromide and sodium dodecyl sulfate-lithium chloride methods [49]. RNA quality was analyzed using a 2100 Bioanalyzer (Agilent Technologies, Santa Clara, CA, USA), and RNA samples with an RNA integrity number ≥ 7 were used for RNA-Seq analyses.

### 5.2. Production of Transcriptome Data

HQ RNA samples from *R. glutinosa* roots, leaves, flowers, and stems were used for transcriptome data production. For RNA-Seq data, the total RNA (2 µg) from each RNA sample was used to construct the Illumina paired-end RNA-Seq library using TruSeq Stranded mRNA LT Sample Prep Kit (Illumina, San Diego, CA, USA) and then sequenced with the Illumina HiSeq X platform (Macrogen, Seoul, Korea). The sequencing amounts from each sample are shown in Appendix A.

For Iso-Seq data, a pooled RNA sample was prepared by pooling the 18 individual RNA samples at equal concentrations and used to construct a library with the SMATer PCR cDNA Synthesis kit & DNA Template prep kit 1.0 (Pacific Biosciences, Menlo Park, CA, USA), which was then sequenced with PacBio Sequel platform (DNALink, Seoul, Korea).

### 5.3. Construction of Unigene Set

Adapter sequences and low-quality reads were removed from Illumina RNA-Seq raw reads using Trimmomatic program (ver. 0.3.9, accessed on 15 June 2022) with default parameters [50]. Thereafter, the bacterial sequence contamination was removed using BBduk program (ver. 38.87, accessed on 15 June 2022) with k-mer 31 parameter [51]. HQ isoform sequences generated by PacBio Iso-Seq and the clean Illumina RNA-Seq reads were hybrid assembled using rnaSPAdes (ver. 3.15.0, accessed on 15 June 2022) with a parameter of normal filter [23] and then duplicated assembled sequences with >90% homology were removed using CD-HIT-EST (ver. 4.8.1, accessed on 15 June 2022) with default parameters to construct a unigene set [52]. CDS and deduced protein sequences were predicted from unigene sequences using TransDecoder (ver. 5.5, accessed on 15 June 2022) with default parameters (http://transdecoder.github.io, accessed on 15 June 2022).

Functional annotations were performed to predict the putative function of unigenes. The homologous sequences of unigenes were searched by similarity analysis (cutoff e-value 1E-5) using DIAMOND program (ver. 0.9.30.131, accessed on 15 June 2022) [53] with the Blastx method against NCBI non-redundant (nr) protein database. GO terms were assigned to unigenes using Blast2GO (ver. 5.2.5, accessed on 15 June 2022) based on the results of similarity analysis [54]. Conserved domains were searched using InterProScan (ver. 5.34–73.0, accessed on 15 June 2022) [55]. Unigenes were assigned to KEGG pathways using the KEGG Automatic Annotation Server (KAAS, https://www.genome.jp/tools/kaas/, accessed on 15 June 2022) with the single-directional best hit method [56]. *Arabidopsis thaliana* genes with homology to unigenes were searched using Blastx analysis with a cutoff e-value of 1e−5 against gene sequences of Araport11 (https://www.arabidopsis.org/index.jsp, accessed on 15 June 2022).

### 5.4. Expression Profiling and DEG Analysis

The clean RNA-Seq reads were mapped on the unigene sequences using Bowtie2 (ver. 2.3.5, http://bowtie-bio.sourceforge.net/bowtie2/index.shtml, accessed on 15 June 2022) [57] and then TPM values were calculated based on RNA-Seq read number mapped on unigene sequences using RSEM program (ver. 1.3.3, accessed on 15 June 2022) with default parameters [58]. The bioconductor package DESeq2 (ver. 1.28.1, accessed on 15 June 2022) was used to identify DEGs between samples and the nbinomWaldTest was applied for statistical significance analysis of DEGs [59]. Genes showing over 2-fold expression changes with an adjusted *p*-value of less than 0.05 were considered as DEGs.

Expression values of individual genes were normalized to Z-scores using the R package heatmap [60], and complete linkage and Euclidean distance were used for heatmap visualization as a measure of similarity. All expression analyses were performed by selecting genes with a TPM value ≥ 5.

### 5.5. GO and KEGG Enrichment Analysis

GO enrichment analysis was performed for the DEGs using Fisher’s exact test with an adjusted *p*-value of 0.05 in Blast2GO (ver. 5.2.5, accessed on 15 June 2022) [54]. KEGG pathway enrichment analysis was performed using fisher.test function in R for the modified Fisher’s exact test (https://david.ncifcrf.gov/content.jsp?file=functional_annotation.html; refer to EASE Score section, accessed on 15 June 2022) and visualized by ggplot function provided by R-package ggplot2.

The gene-rich factor in the GO analysis was calculated as the ratio of the DEGs assigned to the GO term/all genes assigned to that GO term. The gene-rich factor in the KEGG analysis was calculated as the gene ratio of the DEGs assigned to the KEGG pathway/total DEGs.

### 5.6. Analysis of qRT-PCR

One microgram of total RNA extracted from each RNA sample was used to synthesize cDNA using the cDNA EcoDry Premix kit (TaKaRa, Kyoto, Japan). A reaction mixture containing 1 µL of cDNA, 0.5 µL of gene specific primer (10 pmol), and 10 µL of RbTaq qPCR mix with SYBR Green with low Rox (Enzynomix, Daejeon, Korea) was prepared for qRT-PCR. qRT-PCR was performed using Roche Light Cycler 480 II (Roche, Basel, Switzerland) under an initial denaturation of 95 °C for 10 min, followed by 45 cycles of denaturation at 95 °C for 10 s, annealing at 58 °C for 15 s, and extension at 72 °C for 15 s. Expression values were measured based on the cycle threshold (Ct) of each gene. The ΔCt value is the difference between the endogenous reference gene, ELONGATION FACTOR 2, and the target gene Ct value. The relative expression levels were analyzed by calculating the 2–ΔCt values, and significant differences between the samples were determined using the Student’s *t*-test (*p* < 0.05).

## Figures and Tables

**Figure 1 genes-13-01092-f001:**
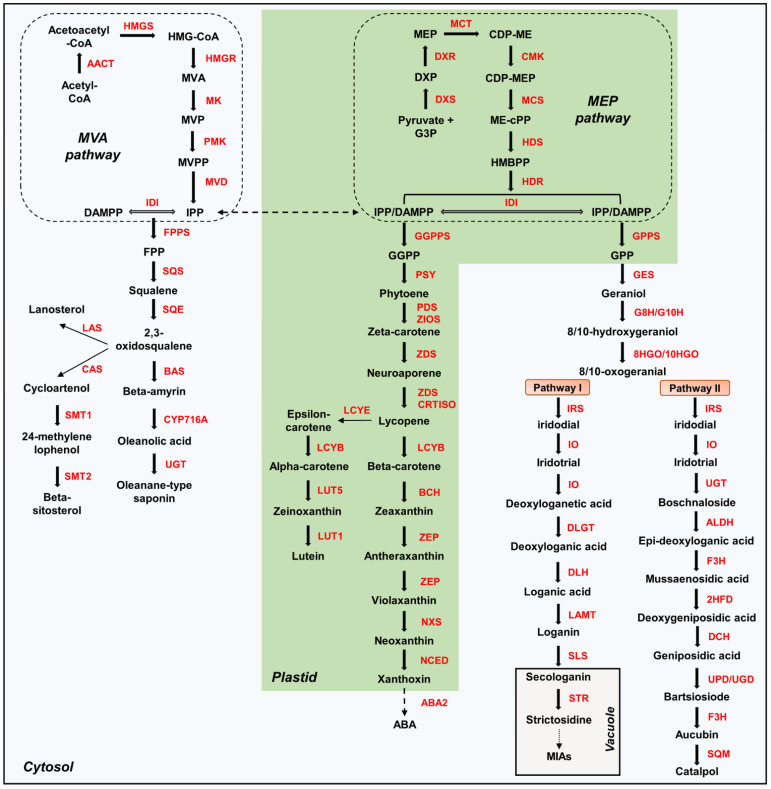
**Terpenoid metabolism in plants. HMG**, 3-hydroxy-3-methylglutaryl; **MVA**, mevalonate; **MVP**, mevalonate phosphate; **MVPP,** mevalonate diphosphate; **IPP**, isopentenyl pyrophosphate; **DAMPP**, dimethylallyl pyrophosphate; **G3P**, glyceraldehyde 3-phosphate; **DXP**, 1-deoxy-D-xylulose 5-phosphate; **MEP**, methylerythritol 4-phosphate; **CDP-ME**, 4-(cytidine 5′-diphospho)-2-C-methyl-D-erythritol; **ME-cPP**, 2-C-methyl-D-erythritol-2,4-cyclodiphosphate; **HMBPP**, 1-hydroxyl-2-methyl-2-butenyl-4 diphosphate; **FPP**, farnesyl diphosphate; **GGPP**, geranylgeranyl diphosphate; **GPP**, geranyl diphosphate; **TIAs**, terpenoid indole alkaloids; **ABA**, abscisic acid; **AACT**, acetyl-CoA acetyltransferase; **HMGS**, hydroxymethylglutaryl-CoA synthase; **HMGR**, 3-hydroxy-3-methylglutaryl-CoA reductase; **MK**, mevalonate kinase; **PMK**, phosphomevalonate kinase; **MVD**, mevalonate diphosphate decarboxylase; **DXS**, 1-deoxy-D-xylulose 5-phosphate synthase; **DXR**, 1-deoxy-D-xylulose 5-phosphate reductoisomerase; **MCT**, 2-C-methyl-D-erythritol 4-phosphate cytidylyltransferase; **CMK**, 4-diphosphocytidyl-2-c-methyl-d-erythritol kinase; **MCS**, 2-C-methyl-D-erythritol 2,4-cyclodiphosphate synthase; **HDS**, 4-hydroxy-3-methylbut-2-enyl diphosphate synthase; **HDR**, 4-hydroxy-3-methylbut-2-enyl diphosphate reductase; **IDI**, isopentenyl-diphosphate delta-isomerase; **FPPS**, farnesyl pyrophosphate synthase; **SQS**, squalene synthase; **SQE**, squalene epoxidase; **BAS**, beta-amyrin synthase; **CYP716A**, beta-amyrin oxidase; **UGT**, UDP-glycosyltransferase; **LAS**, lanosterol synthase; **CAS**, cycloartenol synthase; **SMT1**, sterol 24-C-methyltransferase; **SMT2**, 24-methylenesterol C-methyltransferase; **GGPPS**, geranylgeranyl pyrophosphate synthase; **GPPS**, geranyl pyrophosphate synthase; **GES**, geraniol synthase; **G8H/G10H**, geraniol 8/10-hydroxylase; **8HGO/10HGO**, 8/10-hydroxygeraniol dehydrogenase; **IRS**, iridoid synthase; **IO**, irodoid oxidase; **DLGT**, deoxyloganetic acid glucosyltransferase; **DLH**, deoxyloganic acid hydroxylase; **LAMT**, loganic acid O-methyltransferase; **SLS**, secologanin synthase; **STR**, strictosidine synthase; **ALDH**, aldehyde dehydrogenase; **F3H**, flavanone 3-hydroxylase; **2HFD**, 2-hydroxyisoflavanone dehydratase; **DCH**, deacetoxycephalosporin-C hydroxylase; **UPD**, uroporphyrinogen decarboxylase; **UGD**, UDP-glucuronic acid decarboxylase; **SQM**, squalene monooxygenase; **PSY**, phytoene synthase; **PDS**, phytoene desaturase, **ZIOS**, 15-cis-zeta-carotene isomerase; **ZDS**, zeta-carotene desaturase; **CRTISO**, carotenoid isomerase; **LCYB**, lycopene beta-cyclase; **LCYE**, lycopene epsilon cyclase; **BCH**, beta-carotene 3-hydroxylase; **LUT5**; lutein deficient 5; **LUT1**, lutein deficient 1; **ZEP**, zeaxanthin epoxidase; **NCED**, nine-cis-epoxycarotenoid dioxygenase; **ABA2**, xanthoxin dehydrogenase.

**Figure 2 genes-13-01092-f002:**
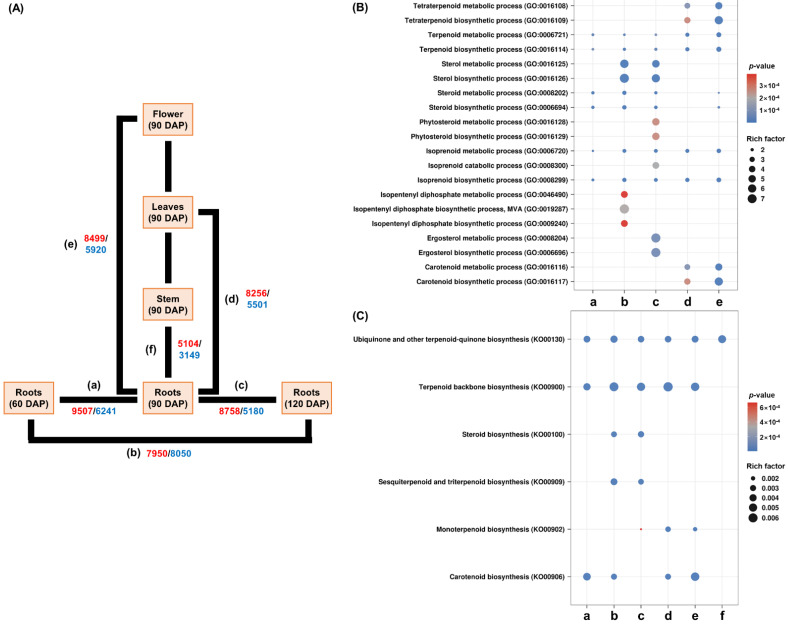
**Analysis of GO and KEGG pathways.** (**A**) Schematic diagram of DEG analysis. DEG analysis between roots’ development and other tissues using six comparative combinations: (**a**) 60 DAP roots vs. 90 DAP roots; (**b**) 60 DAP roots vs. 120 DAP roots; (**c**) 90 DAP roots vs. 120 DAP roots; (**d**) 90 DAP roots vs. 90 DAP leaves; (**e**) 90 DAP roots vs. 90 DAP flowers; and (**f**) 90 DAP roots vs. 90 DAP stems. Red and blue numbers indicate up- and down-regulated DEGs, respectively. (**B**,**C**) The analysis of terpenoid-related GO and KEGG terms using dot plot. The color of the circles represents the *p*-value between each comparison. The size of the circles indicates the DEGs ratio assigned to the GO or KEGG terms/all genes assigned to those GO or KEGG terms, respectively. **DEGs**, differentially expressed genes; **GO**, gene ontology; **KEGG**, Kyoto Encyclopedia of Genes and Genomes.

**Figure 3 genes-13-01092-f003:**
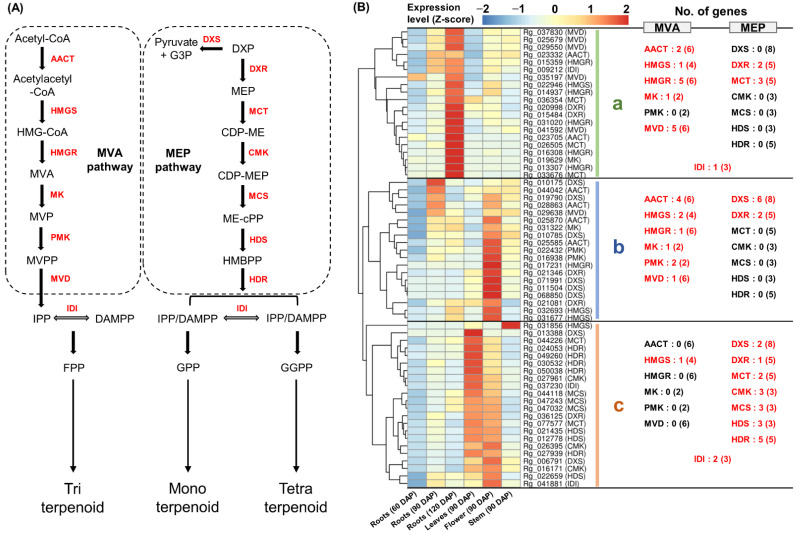
**Expression analysis of genes involved in MVA and MEP biosynthesis.** (**A**) MVA and MEP biosynthetic pathway in plants. (**B**) Heatmap analysis of MVA and MEP biosynthetic genes. (**a**) Genes cluster showing a strong expression in the 120 DAP roots. (**b**) Genes cluster showing a constitutive expression in most tissues. (**c**) Genes cluster showing a strong expression in the leaves and flowers at 90 DAP. Genes highlighted in red indicate genes assigned to each cluster and the number in parentheses is the total number of genes encoding the enzyme expressed in *R. glutinosa* transcriptome. **MVA**, mevalonate phosphate; **MEP**, methylerythritol 4-phosphate.

**Figure 4 genes-13-01092-f004:**
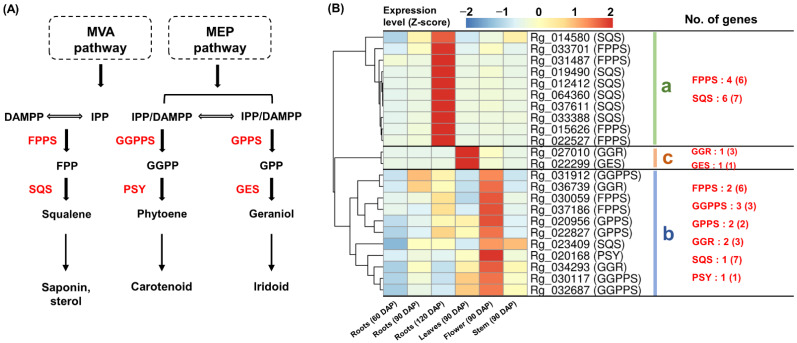
**Expression analysis of TPS encoding genes.** (**A**) The process of squalene, phytoene, and geraniol formation in the terpenoid metabolism. (**B**) Heatmap analysis of TPS encoding genes. (**a**) Genes cluster showing a strong expression in the 120 DAP roots. (**b**) Genes cluster showing a high expression in the 90 DAP flowers. (**c**) Genes cluster showing a specific expression in the 90 DAP leaves. Genes highlighted in red indicate genes assigned to each cluster and the number in parentheses is the total number of genes encoding the enzyme expressed in *R. glutinosa* transcriptome. **TPS,** terpene synthase.

**Figure 5 genes-13-01092-f005:**
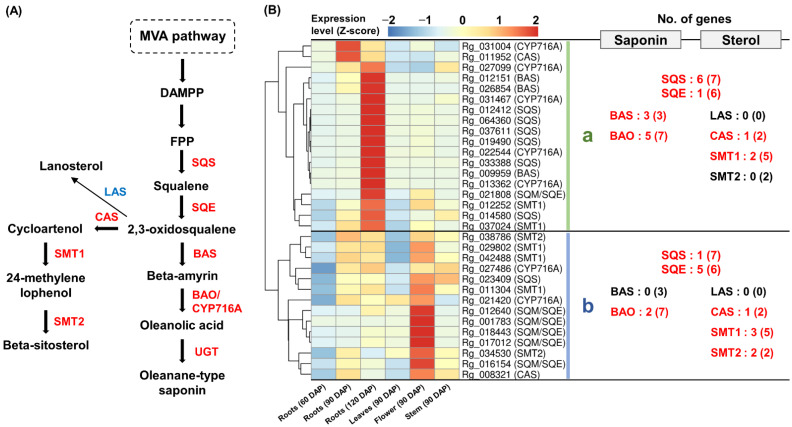
**Expression analysis of genes involved in triterpenoid saponin and sterol.** (**A**) Saponin and beta-sitosterol biosynthetic pathway in plants. Genes that were missing in this study and genes with <5 TPM values are shown in blue. (**B**) Heatmap analysis of genes responsible for saponin and sterol biosynthesis. (**a**) Genes cluster showing a strong expression in the 120 DAP roots. (**b**) Genes cluster showing a constitutive expression in most tissues. Genes highlighted in red indicate genes assigned to each cluster and the number in parentheses is the total number of genes encoding the enzyme expressed in *R. glutinosa* transcriptome. **TPM,** transcripts per million.

**Figure 6 genes-13-01092-f006:**
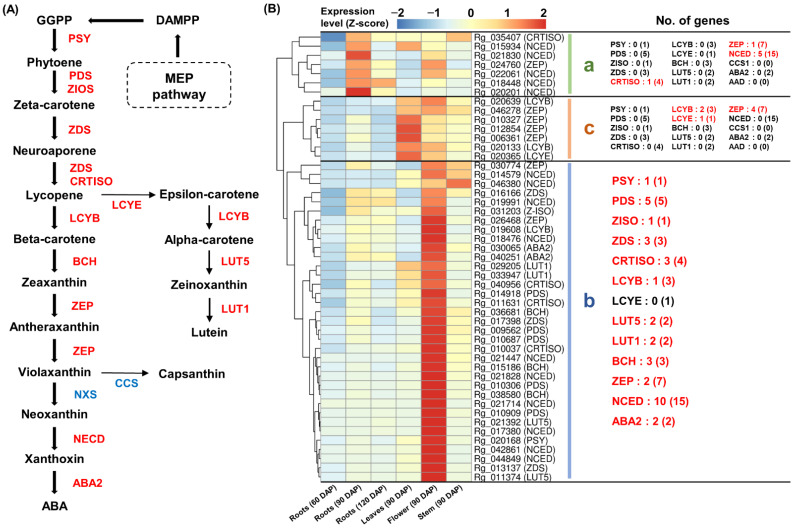
**Expression analysis of carotenoid biosynthesis genes.** (**A**) Carotenoid biosynthetic pathway in plants. Genes that were missing in this study and genes with <5 TPM values are shown in blue. (**B**) Heatmap analysis of genes involved in carotenoid biosynthesis. (**a**) Genes cluster showing a high expression in the 90 DAP roots. (**b**) Genes cluster showing a strong expression in the 90 DAP flowers. (**c**) Genes cluster showing a high expression in the 90 DAP leaves and flowers. Genes highlighted in red indicate genes assigned to each cluster and the number in parentheses is the total number of genes encoding the enzyme expressed in *R. glutinosa* transcriptome. **TPM**, transcripts per million.

**Figure 7 genes-13-01092-f007:**
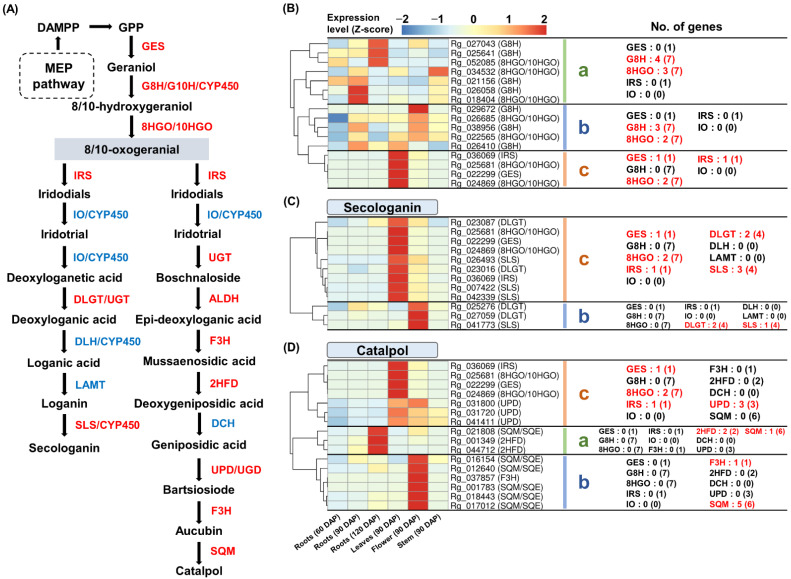
**Expression analysis of two iridoids biosynthesis genes.** (**A**) Secologanin and catalpol biosynthetic pathway in plants. Genes that were missing in this study and genes with <5 TPM values are shown in blue. (**B**–**D**) Heatmap analysis of genes involved in two iridoid pathways. (**a**) Genes cluster showing a high expression in the roots. (**b**) Genes cluster showing a strong expression in the 90 DAP flowers. (**c**) Genes cluster showing a strong expression in the 90 DAP leaves. Genes highlighted in red indicate genes assigned to each cluster and the number in parentheses is the total number of genes encoding the enzyme expressed in *R. glutinosa* transcriptome. **TPM**, transcripts per million.

**Figure 8 genes-13-01092-f008:**
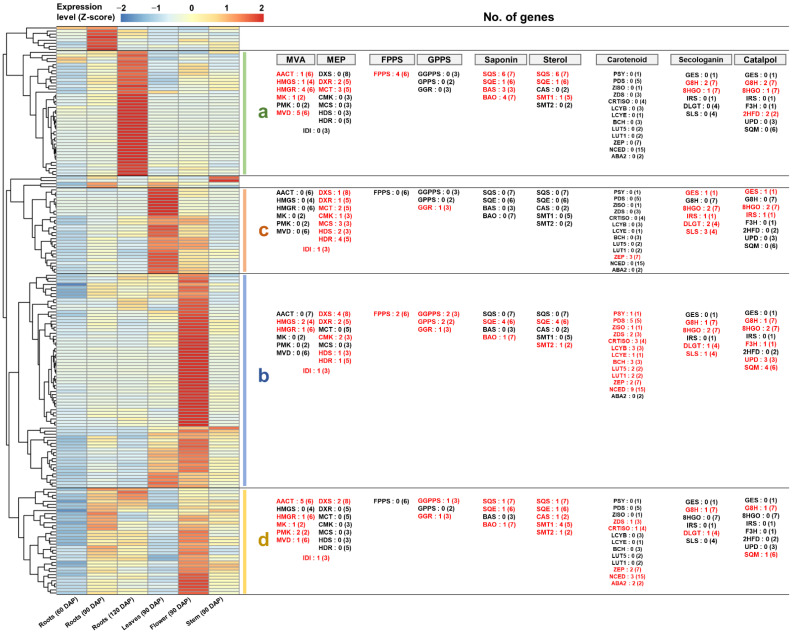
**Comprehensive co-expression analysis of genes responsible for terpenoid metabolism in *R. glutinosa*.** (**a**) Genes cluster showing a high expression in the 120 DAP roots. (**b**) Genes cluster showing a strong expression in the 90 DAP flowers. (**c**) Genes cluster showing a high expression in the 90 DAP leaves. (**d**) Genes cluster showing a constitutive expression in most tissues. Genes highlighted in red indicate genes assigned to each cluster and the number in parentheses is the total number of genes encoding the enzyme expressed in *R. glutinosa* transcriptome.

**Figure 9 genes-13-01092-f009:**
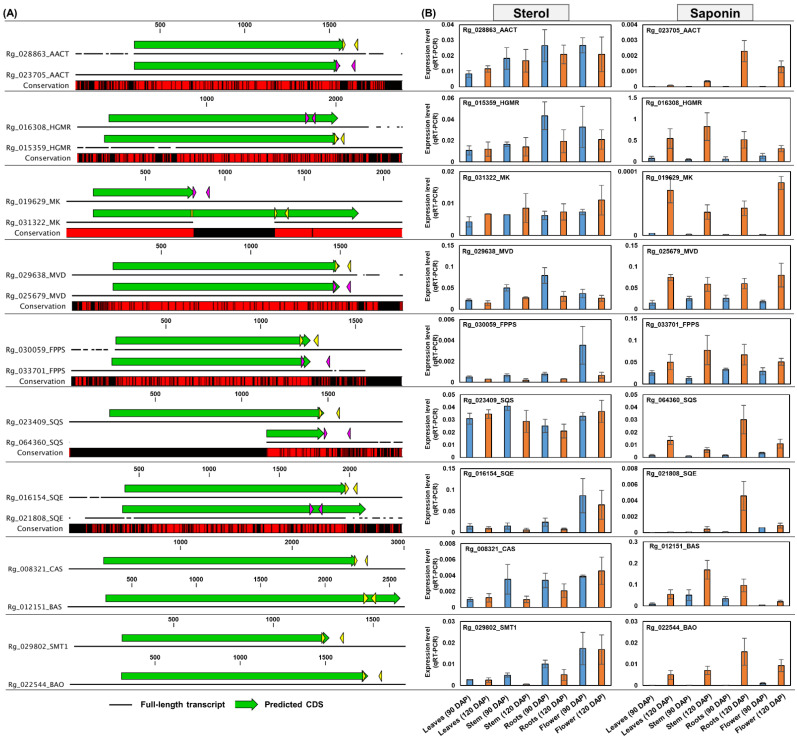
**qRT-PCR analysis of sterol and saponin biosynthesis genes via MVA pathway.** (**A**) Schematic diagram of the nucleotide sequence structure of genes involved in sterol and saponin biosynthesis. Colored arrows indicate specific primer positions for the amplification of each gene. Red regions in conservation boxes indicate identical sequences between homologous genes. (**B**) qRT-PCR analysis. The real-time expression level of each gene was calculated using the delta-CT method and visualized using a bar plot. Asterisks indicate significant differences between 90 DAP and 120 DAP samples (Student’s *t*-test, *p* < 0.05). Error bars represent standard deviations (SD) among three replicates. **MVA**, mevalonate phosphate.

**Figure 10 genes-13-01092-f010:**
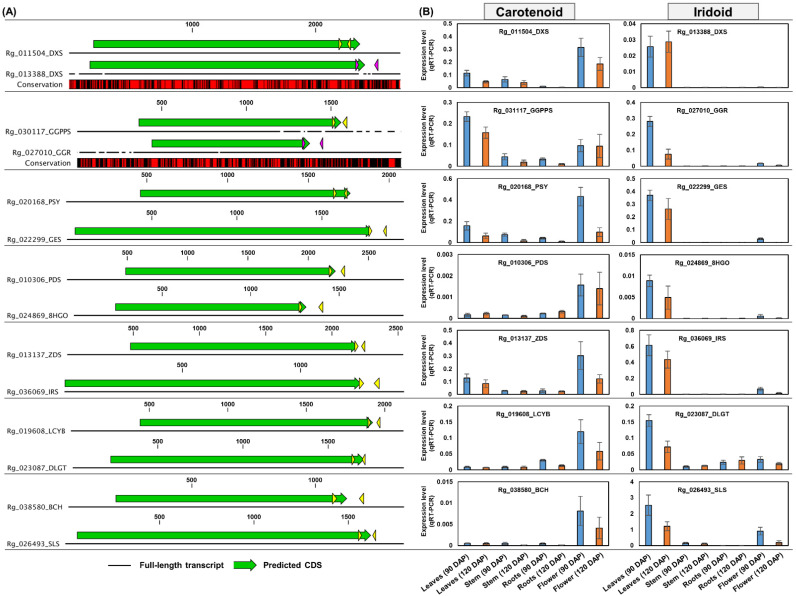
**qRT-PCR analysis of carotenoid and iridoid biosynthesis genes via MEP pathway.** (**A**) Schematic diagram of the nucleotide sequence structure of genes involved in carotenoid and iridoid biosynthesis. Colored arrows indicate specific primer positions for the amplification of each gene. Red regions in conservation boxes indicate identical sequences between homologous genes. (**B**) qRT-PCR analysis. The real expression level of each gene was calculated using the delta-CT method and visualized using a bar plot. Error bars represent standard deviations (SD) among three replicates. **MEP**, methylerythritol 4-phosphate.

**Table 1 genes-13-01092-t001:** Unigene set information generated from *R. glutinosa*.

	Total Unigenes	Predicted CDS	Deduced Protein
Unigene No.	140,335	58,949	58,946
Total length (bp)	149,495,081	57,594,645	19,198,215
Minimum length (bp)	165	255	85
Maximum length (bp)	16,911	16,239	5413
Average length (bp)	1065	977	326
N50 (bp)	1821	1287	429
GC Ratio (%)	39.72	44.14	152
BUSCO ver5	98%	-	-

**Table 2 genes-13-01092-t002:** Functional annotation of 140,355 unigenes.

Database	Program and Parameters	Number of Annotated Genes	Percentage of Total Genes
NCBI nr proteins	DIAMOND, cutoff e-value 1E-5	77,747	55.40%
GO	Blast2GO, default parameters	51,296	36.55%
InterProScan	InterProScan	45,663	32.54%
KEGG	KAAS, SBH method	30,226	21.54%
Araport11	BLASTX, cutoff e-value 1E-5	61,242	43.64%
Total		78,559	55.98%

## Data Availability

The RNA-Seq data used in the present study were submitted to the National Agricultural Biotechnology Information Center (NABIC, https://nabic.rda.go.kr, accessed on 15 June 2022) and are publicly available (Accession No.: NN-7269, NN-6901, NN-6902, NN-6904, NN-6905, NN-6906, NN-6907, NN-6908, NN-6909, NN-6910, NN-6911, NN-6912, NN-6913, NN-6914, NN-6915, NN-6916, NN-6917, NN-6918, and NN-6919).

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
