# Peer review of "Co-Expression Analysis Reveals Differential Expression of Homologous Genes Associated with Specific Terpenoid Biosynthesis in Rehmannia glutinosa"

_genes, 2022, doi:10.3390/genes13061092_

Round 1

Reviewer 1 Report

Although the manuscript presents interesting results, it needs further and major improvement. Therefore, I recommend the authors consider the following point during the revision.

Introduction section

  1. In lines 34-36, I recommend the authors add other pharmacological properties, such as antiviral activities induced by terpenoids against herpes simplex virus.

Materials and Methods section

  1. All experiments should be performed according to proper methodologies. Please add the proper citation for each experiment.
  2. Please provide detailed information about the statistical analyses used for all performed experiments (statistical tests; for example, what tests were used to determine the differences between treatments with test samples and the positive controls, along with information about post-hoc comparison tests and statistical significance). This information is very important. Otherwise, the reported results will not be considered reliable. 

Conclusion section

This section is missing. Please add a conclusion section in which information about the significant findings of this study and its weak points should be highlighted. Moreover, information about further investigations that should be conducted has to be also emphasized. 

Reviewer 2 Report

In this study authors took the transcriptomic approach on R. glutinosa and investigated the expression of genes involved in terpenoid biosynthesis. This study is interesting and set a base for metabolic engineering for the production of terpenoids. Overall, the quality of the manuscript is good. Introduction is very well written; results are nicely explained and discussed. Methods are elaborated.

However, I have two minor suggestions that need to be addressed.

1)      depth of each sample should be mentioned in the methods parts.

2)      This study is based on transcriptomic approach, however I noticed that authors did not provide the reference to access the raw data. Raw data should be deposited to public  database.

Round 2

Reviewer 1 Report

Dear Authors,

Unfortunately, the manuscript has not been significantly improved. All my comments and recommendations have not been addressed. Moreover, you did not include the response to all my comments point by point. Also, no conclusion section supported by the results was added. No detailed statistical analyses were described. The authors should follow the reviewer's comments and recommendations. Unfortunately, the manuscript in its current form is not at the level of publication, since several methodological (especially statistical analyses) errors were detected. Another point is that in the introduction section, I previously recommended adding anti-herpes virus properties induced by terpenoids. This information has not been addressed.  I recommend this reference for this information ''Natural Products-Derived Chemicals: Breaking Barriers to Novel Anti-HSV Drug Development. Viruses. 2020; 12(2):154''.

Finally, during the second revision, please attach your responses to all my comments point by point. This helps me track the changes you have made.